# CAN FOUNDATION MODELS ACTIVELY GATHER INFORMATION IN INTERACTIVE ENVIRONMENTS TO TEST HYPOTHESES?

## ABSTRACT

While problem solving is a standard evaluation task for foundation models, a crucial component of problem solving—actively and strategically gathering information to test hypotheses—has not been closely investigated. To assess the information gathering abilities of foundation models in interactive environments, we introduce a framework in which a model must determine the factors influencing a hidden reward function by iteratively reasoning about its previously gathered information and proposing its next exploratory action to maximize information gain at each step. We implement this framework in both a text-based environment, which offers a tightly controlled setting and enables high-throughput parameter sweeps, and in an embodied 3D environment, which requires addressing complexities of multi-modal interaction more relevant to real-world applications. We further investigate whether approaches such as self-correction and increased inference time improve information gathering efficiency. In a relatively simple task that requires identifying a single rewarding feature, we find that Gemini's information gathering capability is close to optimal. However, when the model must identify a conjunction of rewarding features, performance is suboptimal. The hit in performance is due partly to the model translating task description to a policy and partly to the model's effectiveness in using its in-context memory. Performance is comparable in both text and 3D embodied environments, although imperfect visual object recognition reduces its accuracy in drawing conclusions from gathered information in the 3D embodied case. For single-feature-based rewards, we find that smaller models curiously perform better; for conjunction-based rewards, incorporating self correction into the model improves performance.

## 1 INTRODUCTION

Foundation models have revolutionized natural language processing (NLP) and multi-modal dialogue, achieving remarkable proficiency in understanding and generating human-like text (e.g., Achiam et al., 2023; Gemini Team et al., 2023; Jiang et al., 2024; Reid et al., 2024; Dubey et al., 2024; Dai et al., 2024; Deitke et al., 2024). Their impact extends to applications like machine translation, text summarization, and, increasingly, interactive dialogue agents. As agents, LLMs must not only answer questions but also ask questions and gather information to achieve goals. This type of information gathering is distinct from classic exploration in reinforcement learning (RL), which often uses random exploration policies or prioritizes visitation of novel states (e.g., Burda et al., 2018; Ecoffet et al., 2019; Badia et al., 2020). Targeted information acquisition requires formulating hypotheses about the task and strategically gathering information to test and refine these hypotheses. This hypothesis-driven approach to information gathering remains largely unexamined in foundation models.

To systematically study information-gathering capabilities in foundation models, we need to carefully control the environment (or hypothesis space) in which the model operates. To this end, we propose a parametric class of environments varying in complexity. We realize these formal environments in two distinct implementations: a text-based interaction and an embodied 3D simulation. In the text-based environment, we run experiments at scale, and in embodied 3D environments, we evaluate the effects of additional complications such as partial observability, credit assignment, and scene understanding—complications relevant to real-world applications such as robotics and video games.

More specifically, this paper investigates the capacity and efficiency of foundation models to conduct exploratory behavior within interactive environments in the zero-shot setting, using in-context prompting alone and without requiring task-specific training or fine-tuning. We examine whether foundation models can effectively navigate, adapt, and reason in text-based scenarios that require sequential decision-making and strategic exploration. Furthermore, to assess the generalization of our findings to more complex and multi-modal settings, we evaluate these foundational models in an embodied, vision-based setting that is designed to closely match the text-based environment.

Our experiments with Gemini 1.5 (Reid et al., 2024) reveal significant exploratory capabilities, effective navigation of complex abstract problem spaces, the discovery of novel solutions, and the achievement of predefined objectives with minimal guidance. While performance tends to decrease as environmental complexity increases, such as more complex reward functions or when moving to 3D environments that require visual understanding, exploration efficiency significantly outperforms random baselines. These findings suggest that foundation models possess a latent ability to adaptively gather information in interactive environments, both text-based and embodied, opening exciting avenues for research and application. Overall, this work makes three key contributions:

- **Framework development**: We propose a novel framework for evaluating the directed exploration capabilities of LLMs and VLMs in interactive environments, outlining methodologies for assessment in the zero-shot setting, without the need for fine-tuning or other post-training modifications.

- **Empirical analysis**: We conduct extensive experiments across various environments and tasks, and across several model variants and prompting strategies, to analyze the exploration performance and behaviors of LLMs and VLMs in interactive settings.

- **Insights and implications**: We provide a detailed discussion on the implications of our findings for future research in foundation models and the development of autonomous intelligent agents.

The remainder of this paper is organized as follows: Section 2 reviews related work, Section 3 details our proposed framework and experimental setup, Section 4 presents the results and analysis of our experiments, and Section 5 concludes the paper and discusses potential directions for future research.

## 2 RELATED WORK

**Exploration in RL** Information gathering is related to exploration in RL, which has been studied in particular for tasks with sparse rewards such as Montezuma's Revenge and Pitfall in Atari and the DM-HARD-8 tasks (e.g., Burda et al., 2018; Ecoffet et al., 2019; Badia et al., 2020; Guo et al., 2022; Saade et al., 2023) as well as in unsupervised settings (e.g., Pathak et al., 2017; Guo et al., 2022). These methods commonly derive an "intrinsic" reward from the error of a predictive model (e.g., Pathak et al., 2017; Burda et al., 2018; Guo et al., 2022) or by estimating the density of visited states (e.g., Saade et al., 2023). Badia et al. (2020) use a combination of both of these types of intrinsic rewards and Tam et al. (2022) additionally use pre-trained representations. In contrast, this work relies on the prior knowledge of foundation models from internet-scale pre-training for exploration (e.g., Wang et al., 2023a; Feng et al., 2023; Lu et al., 2024b) rather than using random exploratory actions and intrinsic rewards.

**Foundation models for games** Foundation models have also been used to build agents that play games (e.g., Wang et al., 2023a;b; Feng et al., 2023; Tan et al., 2024), which often involves some form of exploration. Wang et al. (2023a) show that GPT-4 can reach impressive performance in Minecraft by incrementally building a skill library via an "automatic curriculum" stage where GPT-4 is prompted to propose novel tasks. Feng et al. (2023) prompt an LLM to explore an environment and subsequently use the collected experiences for fine-tuning the model. Unlike Wang et al. (2023a) and Feng et al. (2023), Wang et al. (2023b) and Tan et al. (2024) use image observations rather than relying on access to environment internal states. All of these works, however, focus more on improving agent performance rather than performing an explicit, systematic investigation of information gathering for iterative hypothesis testing with foundation models in a controlled, zero-shot setting and in comparison to known optimal policies.

**Exploration with foundation models** Several other works investigate exploration with foundation models, e.g., for text-based environments (Lu et al., 2024b; Huang et al., 2024), reinforcement learning from human feedback (RLHF) (Dwaracherla et al., 2024), and multi-armed bandit problems (Coda-Forno et al., 2023; Krishnamurthy et al., 2024). Unlike Krishnamurthy et al. (2024) and Dwaracherla et al. (2024) and similar to Lu et al. (2024b), this work considers stateful environments. While Lu et al. (2024b) replace components of the exploration method introduced in Ecoffet et al. (2019) with an LLM, this work studies the ability of foundation models to gather information and test hypotheses in-context via zero-shot prompting rather than using LLMs in a more modular fashion. Also adopting more modular approaches, Hu et al. (2024) use foundation models as components in a larger exploration framework and Huang et al. (2024) propose to use a smaller agent to explore the environment and a larger agent to leverage the gathered information.

**Active learning** The field of active learning (Settles, 2009) has long studied how to best acquire data to improve a model's predictions. Active learning methods commonly focus on highly structured data (either i.i.d. or on a graph). In contrast, this work explores efficient knowledge acquisition in more general interactive environments.

**Active embodied question answering** This work studies a similar setup to embodied question answering (EQA) (e.g., Das et al., 2018; Zhu et al., 2023; Majumdar et al., 2024; Ren et al., 2024). Similar to our work, agents in the EQA setting need to actively explore an environment to gather information. Unlike our tasks, however, EQA typically does not involve performing iterative experiments to infer unknown mechanisms in dynamic environments.

**AI for science** Hypothesis generation and testing is central to the scientific method and recent works research the application of foundation models in this broader domain (e.g., Romera-Paredes et al., 2023; Trinh et al., 2024; Lu et al., 2024a). Such works typically use foundation models in a highly structured protocol designed for the fixed domain in question. In contrast, this work explores the capabilities of base foundation models as the complexity of the domain varies.

## 3 ENVIRONMENTS FOR EVALUATING EXPLORATION

Existing RL environments (e.g., Todorov et al., 2012; Brockman et al., 2016; Tassa et al., 2018) often conflate exploration with other aspects of agent performance, making it difficult to isolate and assess a model's inherent exploratory capabilities. Such aspects include sparse or deceptive rewards and noisy, non-stationary, or multi-agent environments. We therefore designed a suite of environments that allows us to systematically disentangle and control the factors influencing exploration.

We begin with simplified, text-based environments where the world is abstractly represented. This allows us to assess the model's ability to explore and update its beliefs based solely on textual information, without the complexities of visual perception and motor control. We then progress to a more naturalistic setting, evaluating the model on video inputs from agents acting in an embodied 3D environment. This transition allows us to assess the generalizability of exploratory behaviors to a setting more reflective of real-world applications.

### 3.1 TEXT-BASED ENVIRONMENT

Drawing inspiration from well-studied cognitive tasks explored in Shepard et al. (1961), we adapt a similar structure to investigate information gathering strategies in foundation models. Our tasks involve presenting the model with multiple objects, only a few of which lead to a reward. This mirrors the sparse reward setting common in RL. Each object possesses two or three properties ("features"), such as color, shape, and texture. A specific property or a combination of two properties determines whether an object yields a reward.

For instance, the model might be presented with numerous objects, but only red objects are rewarding. In this case, the model is informed that a single property (either color or shape) determines reward. The optimal strategy then involves exploring unique combinations of color and shape with each attempt. If a "red book" yields no reward, the model should eliminate both "red" and "book" from further consideration.

To assess the capabilities of foundation models in this environment, we modulate task difficulty by adjusting two key aspects: the number of distinct colors (increasing the cognitive load) and the

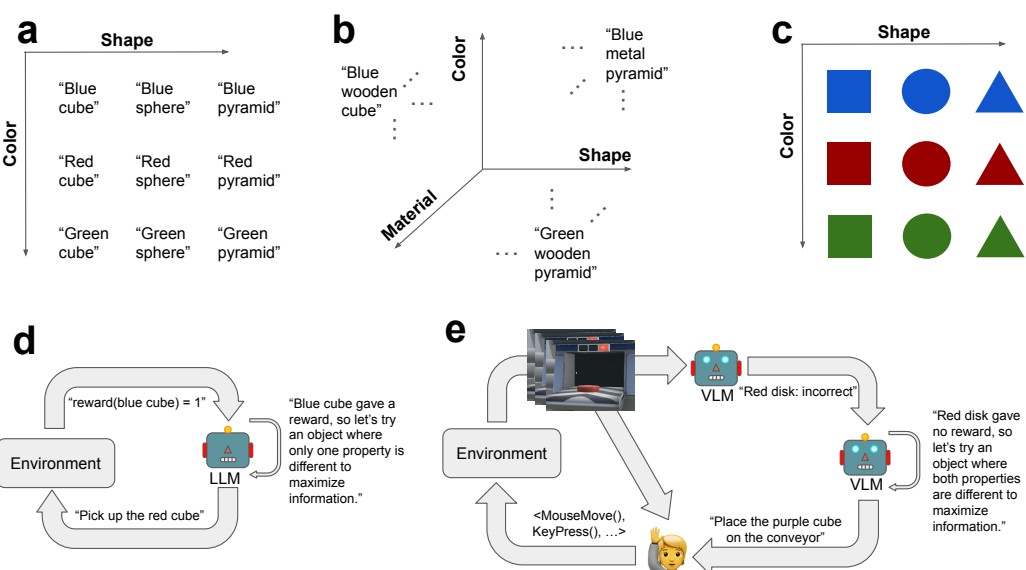

Figure 1: Text-based and embodied exploration experiment designs. (a) Example setup for text environment with single-feature reward function. (b) Example experimental setup for text environment with conjunction reward function. (c) Schematic example setup for 3D embodied environment. The setup mirrors the single-feature text environment experiments in terms of objects present and the reward function, but objects are placed randomly throughout a room as 3D assets in a simulation, rather than provided to the model as a textual list. (d) Active exploration in text environments. (e) Active exploration in a 3D embodied environment.

complexity of the reward function. Reward functions can be based on a single property (**single-feature tasks**) or a conjunction of two properties (**conjunction tasks**). See Figure 1 for a visualization of the tasks.

## 3.2 CONSTRUCTION LAB ENVIRONMENTS

To further evaluate the foundation models in a 3D embodied environment, we implement an analogous task to the text-based environment in a factory-style simulation called Construction Lab. Construction Lab was introduced in **[reference-anonymized]** as a simulation environment that includes both game-like mechanics and simplified but non-trivial object manipulation and physical reasoning.

In this work, we focus on a task that requires the player to operate a simple machine called the Exchanger. The Exchanger requires objects with specific properties to be placed on an input conveyor belt (Figure 2). If an object matches the requirement, the input is consumed, a green light shows for a few seconds, and an output object is produced on an output belt. If the object is invalid, the machine rejects it by reversing the input belt and a red error light is activated. No cues are provided regarding the correct input object required, and thus the task entails determining what the correct object properties are through trial and error, observing how the machine responds to input objects, and drawing appropriate inferences.

Through the use of this 3D, visually rich environment that mirrors the challenges of the text-based environment, we are able to disentangle the effects of visual complexity and imperfect image understanding from the patterns of performance related to reasoning from language alone.

## 4 EXPERIMENTS

Our experiments aim to address the following key questions: (a) How does the complexity of the environment affect the exploration performance of foundation models? See results in §4.1. (b) How does the size of a foundation model impact its exploration capabilities? See results in §4.1.

Figure 2: The Construction Lab environment. The player is tasked with picking up objects of different shapes and colors and placing them on the input conveyor to a machine. **Left:** The agent uses a blue laser beam to pick up objects. **Center:** Result of a correct object placement. **Right:** Result of an incorrect object placement.

(c) How do different approaches, such as self-correction, few-shot generalization, and in-context exploration with long context windows, improve exploration in foundation models? Results in §4.2. (d) What factors hinder model performance in exploration tasks? Is it primarily reasoning ability or the cognitive load imposed by the task? Results in §4.3. (e) Do the performance trends observed in text-based environments generalize to 3D embodied environments? Results found in §4.4.

**Baselines** We compare to two baselines: Optimal Baseline: This baseline represents an upper bound on exploration performance. It selects actions that maximize information gain at each step by reducing uncertainty about the target property. Random Baseline: This baseline establishes a lower bound by choosing actions randomly. We further differentiate the baseline into two variants: Random without Replacement: This variant assumes perfect memory of past actions, making it a more challenging baseline. It's akin to an agent with infinite cognitive capacity, never repeating the same action. Random with Replacement: This variant allows for repeated selection of the same actions, reflecting a more realistic scenario with limited memory.

To illustrate the optimal strategy, consider a single-feature task where color is the only determinant of reward, and the target property is "red." An optimal strategy would systematically test unique color-shape combinations. For instance, if the agent tries "blue toy" and receives no reward, it can rule out both "blue" and "toy". Subsequently, "yellow sphere" would be an optimal choice, as it provides maximum information gain, unlike "blue sphere" which offers no new information about the target property. Once "red box" is tried and yields a reward, the optimal strategy would then focus on isolating whether "red" or "box" is the target property by changing one factor at a time.

In contrast, the random baseline in a pick-up object task might simply select any previously unpicked object to discover its reward. This highlights the contrast between strategic information gain (optimal baseline) and undirected exploration (random baseline).

**Foundation Models** We evaluate on state-of-the-art (SOTA) foundation models of two different sizes, Gemini 1.5 Pro and Gemini 1.5 Flash (Reid et al., 2024).

**Evaluation** Effective exploration necessitates two key abilities. 1) Deciding what to explore: agents must strategically select actions to gather information about the environment. 2) Reasoning and updating beliefs: agents must interpret observations, update their understanding of the world, and make inferences based on new evidence. To assess these abilities, we evaluate the following: (a) Exploration efficiency: We assess how quickly models gather sufficient information (in an information theoretic sense) to identify the specific properties associated with rewards, assuming perfect reasoning. This measures the model's active exploration capabilities independent of its ability to draw conclusions from its observations. (b) Property identification accuracy: We assess the accuracy with which the model identifies the rewarding property or properties based on its observations collected during a fixed budget of exploration steps. This measures both the model's ability to collect observations efficiently and its ability to reason and draw conclusions from those observations.

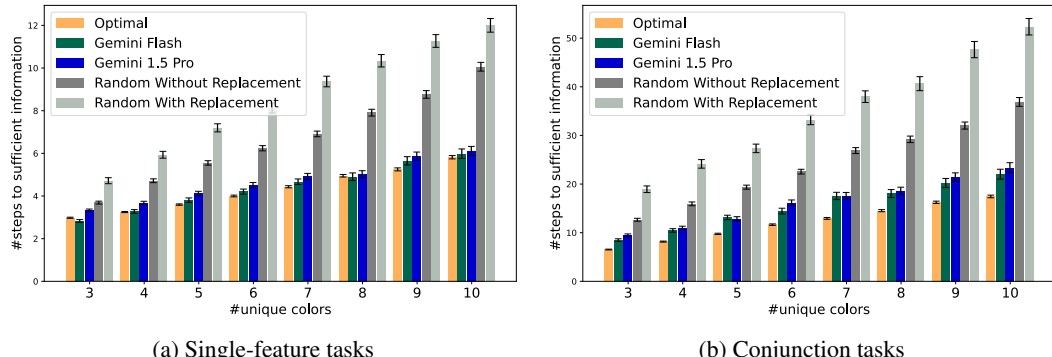

(a) Single-feature tasks    (b) Conjunction tasks

Figure 3: Number of steps to gain sufficient information (lower is better) in the text environment with one and two causal factors. Comparisons between Gemini's (Pro and Flash) exploration policy compared to the optimal policy and random policy, in terms of average number of steps needed to gather sufficient information to discern the rewarding feature. Error bars represent standard error of the mean.

## 4.1 EFFECT OF ENVIRONMENTAL COMPLEXITY ON EXPLORATION

We first examine the effect of varying the amount and type of environmental complexity on strategic exploration and performance. We evaluate the 2 Gemini models' performance in both single-feature and conjunction tasks. We benchmark against an optimal baseline (upper bound) and random baselines with and without replacement (lower bounds assuming perfect memory and no memory, respectively), and evaluated the optimal and random policies on 1000 episodes and the Gemini models on 200 episodes.

**Impact of reward function complexity** To investigate how reward function complexity affects exploration efficiency, we designed tasks where reward is determined by either a single feature (like "red" or "square") or a conjunction of two features (e.g., "red" and "square"). The latter requires the agent to reason about multiple properties to identify the reward-relevant combination.

Figure 3a compares performance in the single-feature task, while Figure 3b shows performance in the conjunction task. We observed that Gemini's performance relative to the optimal baseline declines as reward function complexity increases (from single to conjunction of features). This degradation suggests that in more complex tasks, increased reasoning complexity hinders efficient information gathering.

**Impact of in-context memory load** In LLMs, in-context memory acts as a form of working memory, and as the volume of information stored in this memory grows, so does the cognitive load on the model, potentially affecting its processing and response generation capabilities. To investigate this relationship, we vary the number of unique colors or shapes in the environment, forcing the agent to utilize its in-context memory more efficiently. By evaluating exploration efficiency as the number of unique colors increases in both single-feature and conjunction tasks, we can assess the impact of cognitive load on performance. Figure 3 shows the average number of steps required for the model to gather sufficient information in both single-feature and conjunction tasks. In both tasks, the model significantly outperforms the two random baselines, and is very close to the optimal baseline.

In the single-feature task, both Gemini 1.5 Pro and Gemini Flash perform comparably to the optimal baseline, even as the number of unique colors increases. However, in the conjunction task, while still significantly outperforming the random baselines, performance degrades slightly as the number of unique colors increases. This indicates that increased cognitive load negatively impacts the model's exploration efficiency. In both single-feature and conjunction tasks, Gemini 1.5 Pro and Gemini 1.5 Flash performed comparably for individual numbers of colors. Interestingly, however, Gemini 1.5 Flash was found to outperform the Pro model in the single-factor condition when considering all numbers of colors taken together (see statistical analysis below).

We also find that Gemini effectively exploits information gathered over time, efficiently updating its understanding of the environment. See Section A.1 for details.

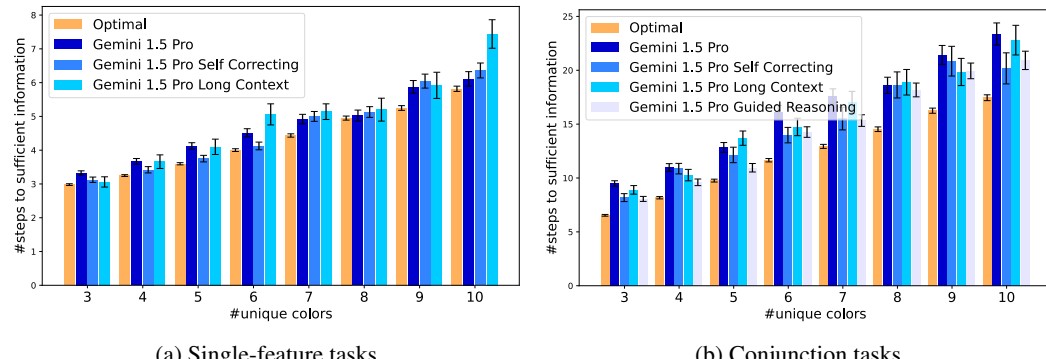

(a) Single-feature tasks   (b) Conjunction tasks

Figure 4: Number of steps to gain sufficient information (lower is better) in the text environment with one and two causal factors. Comparisons between Gemini's exploration policy with and without self-correction and long context, as well as Gemini Flash. Error bars represent standard error of the mean.

## 4.2 EFFECTS OF PROMPTING AND CONTEXT LENGTH

We also investigate whether existing methods for enhancing reasoning abilities in LLMs can improve performance on our tasks. We evaluate the impact of two techniques: 1) Self-correction: allowing the LLM to critique and revise its own reasoning, and 2) Increased inference time: providing the model with its previous response and additional time to reason.

**Self-correction** Building upon previous work exploring the self-correction capabilities of LLMs in mathematical problem-solving (e.g., Huang et al. (2023)), we investigate whether LLMs can self-correct within our task framework. We adapt self-correction prompts similar to those used in Zheng et al. (2024), prompting the model at each step to revise its output, including its reasoning traces. This process allows the model to verify its chosen action and make corrections if necessary. Figure 4 presents the results of this process for both single-feature task and conjunction task scenarios using Gemini 1.5 Pro, and the results for the Flash model is found in Figure 7 in the appendix. In single-feature tasks, it improves performance with up to 6 unique colors, but its benefits diminish with a larger number of colors. Notably, self-correction appears more effective in more complex conjunction tasks, either performing comparably, or slightly outperforming the base model.

**Longer Inference Time** We also explored whether encouraging the model to engage in more deliberate reasoning, by providing it with additional context, could improve performance. Instead of simply providing the model with its previous observations (actions and outcomes), we also included its reasoning traces, explaining why it selected previous actions. This approach, while increasing inference time, allows the model to reflect on its own chain-of-thought, potentially leading to improved reasoning and better decision-making. Figure 4 presents the results of incorporating the model's reasoning traces. In both single-feature and conjunction tasks, this approach, which encourages more deliberate reasoning, yields comparable performance to the baseline approach without reasoning traces.

**Statistical comparisons across models and approaches** In order to quantitatively assess whether model size or prompting method affected exploration efficiency, controlling for the number of colors, we performed analyses of covariance for the single-feature and conjunction tasks separately. First we compared Gemini 1.5 Pro to Gemini Flash: in the single-feature tasks Gemini Flash was significantly better ($F(1, 7649) = 6.1$, $p < 0.05$), in the conjunction tasks there is no significant difference. Second we compared the variants of each model to the base model: in the single-feature task there were no significant differences, however for Gemini 1.5 Pro in the conjunction task we found that the guided reasoning and self-correcting variants were significantly better than the base model ($F(3, 5514) = 5.3$ and $3.0$ respectively, $p < 0.05$ corrected for multiple comparisons). This suggests that when the reward function is simple, it may help to have a smaller model and simple/short reasoning process. Conversely, when the reward function is more complex, it may help to have a more iterative reasoning process.

### 4.3 Impact of Reasoning and In-Context Memory

Efficient exploration requires agents to reason effectively about exploration strategies and maintain a working memory of untried options. However, we observed suboptimal exploration performance by our agent on the conjunction task (Figure 3). To investigate this performance gap, we sought to disentangle the effects of reasoning and memory limitations. We provided Gemini with a guided reasoning strategy, eliminating the need for the agent to independently derive the optimal approach, refer to Table 4 in the appendix for detailed prompts).

Figure 4 (b) demonstrates a clear and consistent performance improvement with guided reasoning, indicating that reasoning challenges contribute significantly to the performance gap. While imperfect adherence to the guided strategy could be a factor, the gap between the guided reasoning model and the optimal policy widens as the number of unique colors increases. This strongly suggests that memory constraints also play a crucial role in limiting the performance of the standard Gemini policy.

### 4.4 Exploration in 3D embodied environments

#### 4.4.1 Task setup

A number of additional challenges must be addressed when performing this exploration task in a 3D embodied environment. First, the agent must assess both the current state of the environment and the consequences of any actions taken through vision. Second, the agent requires a motor control module to execute exploratory actions. We use Gemini 1.5 Pro's multi-modal functionality to ingest video input from Construction Lab sub-sampled to 1.5 Hz and 320 x 240 resolution. To disentangle vision and reasoning performance from translation of natural language instructions into a complex keyboard-and-mouse action space, we adopt a setup in which instructions are provided to a human actor who performs the exploratory actions online.

We assess Gemini 1.5 Pro's ability to generate these exploratory instructions by comparing against an optimal and random baseline, mirroring those in the text environment. The optimal strategy was performed by a single human performing the task according to an optimal policy that maximally reduced uncertainly about the correct property. The random strategy was performed according to a policy that selects a random object from the room at each step, with replacement.

As running the 3D environment and using human actors in the loop reduces experimental throughput, we limit ourselves to a single level of environment and reward function complexity. We choose the condition with 3 colors and 1 causal factor, as conditions with more colors had significant visual clutter. Each task is randomly generated as follows: at the beginning of the episode, 3 unique colors and 3 unique shapes are randomly selected from 6 colors and 5 shapes, and objects with each shape-color combination are placed in random locations in the environment, for a total of 9 objects. One property, either a shape or a color, is randomly selected as the correct property, for a total of 3 correct objects. The player and the Exchanger machine with input and output conveyor are likewise placed randomly in the room. A gameplay episode ends when either all 3 correct objects are placed on the input conveyor or 2 minutes have elapsed.

#### 4.4.2 Gemini-based agent

The Gemini agent is implemented as follows: every 10 seconds, the model is fed the most recent 100 video frames (or 67 seconds) of gameplay and queried in two stages, during which gameplay is paused for the human actor. We implement a two-stage procedure with a vision stage and a reasoning stage, which we found improves accuracy for each stage compared with running both together. In the first stage, Gemini is asked to list, for every object placed on the input conveyor, the timestamp at which it was placed, its color and shape, and whether it was correct or not (as indicated by a red or green light on the machine). In the second stage, Gemini is provided the output of the first stage (subsequent video frames and list of objects placed with their reward values) and prompted to select a next exploratory action to maximize information gain, similar to the text environment. The human actor is provided only with the command generated by the second stage, such as "place the red cube on the conveyor."

We process each video trajectory as follows, regardless of how the exploration instructions were generated. We truncate the video to include only the first 4 object attempts, similar to the text

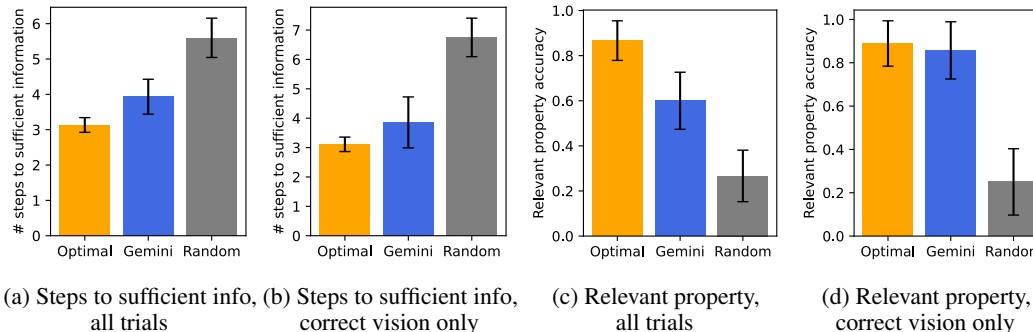

(a) Steps to sufficient info, all trials    (b) Steps to sufficient info, correct vision only    (c) Relevant property, all trials    (d) Relevant property, correct vision only

Figure 5: Performance metrics for 3D exploration task. (a) Mean number of exploration steps (objects placed on the conveyor) before sufficient information is available to determine the correct factor. (b) Same as (a), except with trials in which Gemini made visual errors filtered out. (c) Accuracy of the model in determining the correct rewarding feature. (d) Same as (c), except with trials in which Gemini made visual errors filtered out. Error bars represent standard error of the mean.

environment. Gemini is then called on the truncated video in three steps: vision, reasoning, and generalization. In the vision step, it is asked to list all the objects placed on the conveyor and whether they were correct, similar to the vision step in the exploration policy. In the reasoning step, it is asked to deduce the correct object property based on its observations. See Appendix B for specific prompts used.

### 4.4.3 EVALUATION

We use the same two performance metrics used in the text environment to evaluate different aspects of performance for each agent type: relevant property accuracy, and number of objects until sufficient information is acquired to determine the correct property, assuming perfect reasoning. We also record the number of vision errors made by the VLM when listing objects in the full video, defined as misclassifying the shape, color, or correctness of an object placed on the conveyor, or omitting mention of an object placed on the conveyor (reported in Appendix, Figure 8). Because internal game states are not exposed in our experiments, we use manual human annotation of video trajectories to collect the above metrics and error counts. We collect a total of 15 trajectories for each agent type.

### 4.4.4 RESULTS

In the exploration efficiency metric, we see the same trends in the results for the 3D embodied environment as for the text environment, with Gemini's exploration efficiency significantly out-performing the random baseline and approaching the optimal baseline (Figure 5a). The absolute performance matches that seen in the text experiments, within the margin of error: a mean of 2 steps for Gemini and the optimal baseline, and 4 steps for the random baseline. These results suggest that the additional complexity of an imperfect vision system and partially observed environment state are not significant limitations in generalizing directed exploration capabilities to embodied 3D environments. In the accuracy metric (Figure 5c), the picture is more nuanced. For relevant property accuracy, the difference between performance with the Gemini agent and the random agent was not statistically significant ($p > 0.05$, paired sample t-test).

This result is interesting because VLM vision is also necessary for the exploration phase, where there was no discrepancy in performance. A likely reason for this is that the iterative nature of the exploration task makes it robust to occasional errors. Because the model must re-list all objects placed at each step, chance errors made during one step do not propagate to later steps.

To probe the reason for the gap in accuracy performance, we also computed results where we filtered out trajectories in which the vision step made an error (Figure 5b,d). In these results, accuracies for the Gemini and optimal agents are nearly identical and their differences with the random agent are statistically significant ($p < 0.05$, two sample t-test). These results suggest that errors in the vision step, rather than reasoning or exploration, are responsible for the relatively reduced accuracy in the Gemini agent condition. Investigating relative numbers of errors, there appear to be more

vision errors in the Gemini agent condition than in the optimal or random agent conditions (Figure 8), although these differences aren't statistically significant.

Taken together, results in the Construction Lab show that the directed exploration capabilities of foundation models robustly generalize from text-based environments to embodied 3D environments, though overall accuracy of the system is somewhat reduced by imperfect performance of the VLM's object and action recognition in videos. This indicates that the challenges of multi-modal reasoning from realistic simulated video could be addressed by focusing on the vision and action recognition capabilities of foundation models separately from their reasoning capabilities.

## 5 DISCUSSION AND CONCLUSION

We introduce a framework for evaluating how effectively foundation models can explore new interactive environments and predict hidden reward functions. Our analysis identifies distinct challenges in generating and executing strategic exploratory actions, offering potential solutions.

In a text-based implementation, we evaluate leading foundation models across varying environment and reward complexities. We find that exploration efficiency remains relatively constant compared to an optimal baseline, even as complexity increases. However, performance declines with reward functions based on multiple features, partly due to limitations in policy translation and in-context memory use. (See Section 4.3 for details).

Statistical analysis reveals that Gemini Flash excels with simpler reward functions, while Gemini Pro with self-correction performs better on those with multiple factors. This suggests a potential trade-off between model size/reasoning complexity and reward function complexity. Further research is needed to understand how iterative reasoning influences effective exploration strategies.

In 3D environments, Gemini 1.5 Pro achieves near-optimal exploration efficiency, mirroring its performance in text-based settings. However, accurate interpretation of visual observations remains a challenge due to limitations in the vision system. Improving visual accuracy, potentially through fine-tuning, is important for achieving comparable performance in 3D embodied environments.

The strong performance of foundation models in our exploration tasks motivates further research with more complex environments and methods for improving visual perception. Future directions include replacing the human actor in our 3D setup with language-conditioned agents (Abi Raad et al., 2024; Wang et al., 2023a;b; Feng et al., 2023; Tan et al., 2024) or utilizing real-world footage from head-mounted cameras to further enhance exploration capabilities.

We are excited by the potential of foundation models to autonomously explore and test hypotheses in interactive environments, a crucial aspect of human learning and scientific progress. We anticipate further research in this promising area.

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
