

Figure 6: **Knowledge exploitation efficiency**: We examined how quickly our models exploited the information they gathered. We defined a scoring function that assessed how well our models predictions matched the ground-truth and evaluated the models' predictions after every step. Results are from conjunction tasks, using Gemini 1.5 Pro.

# A    ADDITIONAL RESULTS

Here we present some additional results.

## A.1    KNOWLEDGE EXPLOITATION EFFICIENCY

We examined how well Gemini exploited the information they gathered through exploration (see Figure 6). The scoring function resolves to 1 if every one of the words in a target string (defined

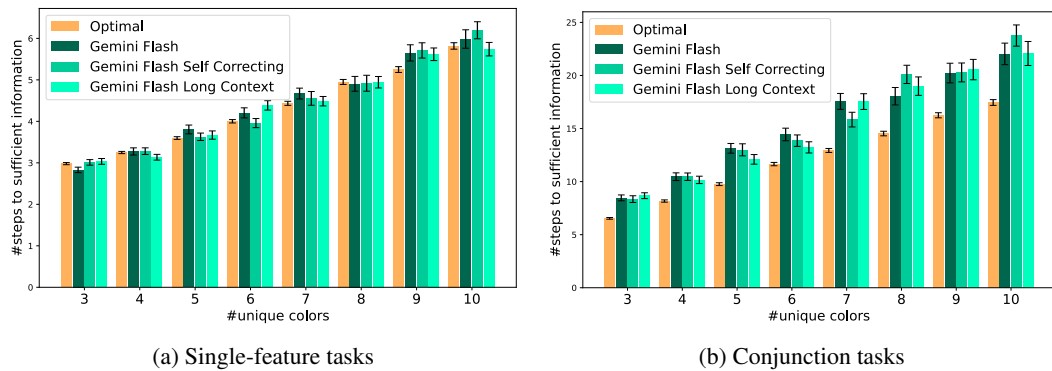

(a) Single-feature tasks  (b) Conjunction tasks

Figure 7: Number of steps to gain sufficient information (lower is better) in the text environment with one and two causal factors. Comparisons between Gemini 1.5 Flash's exploration policy with and without self-correction and long context. Error bars represent standard error of the mean.

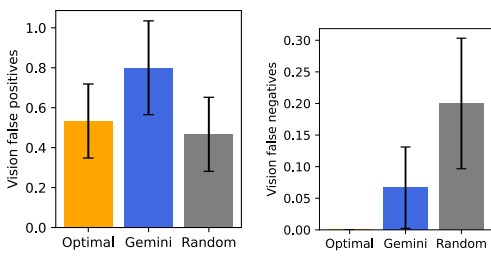

(a) Vision false positives  (b) Vision false negatives

Figure 8: Vision error rates for Construction Lab experiments.

by the correct property or conjunction of properties) is present in the model's answer string, and resolves to 0 otherwise. For example, for a reward function defined by the conjunction of factors "Blue" and "Cylinder", the target string would be "Blue Cylinder". In this case, a model output of "* **WINNING COMBINATION:** COLOR, SHAPE (Blue, Cylinder)" would resolve to 1.

## A.2 RESULTS ON GEMINI 1.5 FLASH

Results on Gemini 1.t Flash is found in Figure 7. There is no clear indication that self-correction or long context impacts performance.

## A.3 VISUAL ERRORS

We identified and classified visual errors that Gemini made, and the results are found in Figure 8.

## B PROMPTS

Prompts used for the foundation models. Prompts are provided verbatim, with the exception of newlines added to fit the text within the table boundaries.

| Task | Prompt |
|------|--------|
| Text Environment Single Factor Task | You are playing a text-based game. Your goal is to discover how to earn rewards.

Game Rules:
- Find as what factors lead to reward as quickly as possible.
- You cannot pick up the same object twice.
- There are objects with different colors, shapes, and textures.
- Picking up an object gives you a reward (either 0 or 1).
- The same object always gives the same reward.
- A specific property, such as a particular color OR shape leads to a reward, find out what it is.

The reward is binary (0 or 1). Only one factor, either color OR shape, determines the reward.  Within the relevant factor, only ONE specific color or ONE specific shape will yield a reward of 1.
If the chosen object matches this correct color (when color is the factor) OR this correct shape (when shape is the factor), the reward is 1. Otherwise, the reward is 0.

{scene_description}

Important: You have VERY FEW turns left. Choose your next action carefully to maximize information.

You are an AI agent designed for thoughtful exploration. Your mission is to navigate and learn within a given environment by performing actions and observing the outcomes.  Operate as a scientist, carefully considering your actions and their consequences.

Exploration Cycle:

- **Action**: Choose an action to perform within the environment. Initially, this may involve random exploration to gain basic understanding.
- **Observe**: Observe the result of your action. This includes any changes to the environment and any rewards or penalties received.
- **Record**: Maintain a detailed log of your actions, observations, and received rewards.
- **Review**: Periodically, pause to explicitly review your action history and the corresponding outcomes. Analyze this data to identify patterns, trends, and potential cause-and-effect relationships.
- **Reason**: Based on your review, reason about the environment.
  What hypotheses can you form about the underlying rules or structure of the environment?
   Are there any actions that seem particularly promising or detrimental?
   Do certain sequences of actions lead to predictable outcomes?
- **Hypothesize**: Clearly state your current hypothesis about the most effective strategy for exploration or achieving a goal.
- **Plan**: Based on your reasoning and hypothesis, plan your next action or sequence of actions. Aim to test your hypothesis and gather more information.


{action_reward_description}

Respond with this format, please be specific about the object:

* Action: pick up <colored> <object>
* Stop: <YES> or <NO>
*
* Which factor influence reward? <COLOR> or <SHAPE>  or <UNSURE>
* WINNING COMBINATION: <State the specific color or shape that leads to reward>
Explain your reasoning thoroughly. Don't just guess! Each turn is precious. |

Table 1: In-context prompt used for the text environments for the single-feature tasks.

| Task | Prompt |
|------|--------|
| Text Environment Multi Factor Task | You are playing a text-based game. Your goal is to discover how to earn rewards.

Game Rules:

- Find as what factors lead to reward as quickly as possible.
- You cannot pick up the same object twice.
- There are objects with different colors, shapes, and textures.
- Picking up an object gives you a reward (either 0 or 1).
- The same object always gives the same reward.
- A specific combination of properties, such as color and shape, shape and texture, or color and texture, leads to a reward. Determine the correct combination.

The reward is binary (0 or 1). Only ONE specific combination of 2 factors will yield a reward of 1.

If the chosen object matches this correct color and shape (when color and shape are the factors), the reward is 1.

Otherwise, the reward is 0. Therefore if an object has reward 0, then all the 3 combinations of 2 factors do not yield reward.

{scene_description}

Important: You have VERY FEW turns left. Choose your next action carefully to maximize information.

You are an AI agent designed for thoughtful exploration. Your mission is to navigate and learn within a given environment by performing actions and observing the outcomes. Operate as a scientist, carefully considering your actions and their consequences.

Exploration Cycle:

- **Action**: Choose an action to perform within the environment. Initially, this may involve random exploration to gain basic understanding.
- **Observe**: Observe the result of your action. This includes any changes to the environment and any rewards or penalties received.
- **Record**: Maintain a detailed log of your actions, observations, and received rewards.
- **Review**: Periodically, pause to explicitly review your action history and the corresponding outcomes. Analyze this data to identify patterns, trends, and potential cause-and-effect relationships.
- **Reason**: Based on your review, reason about the environment.
  What hypotheses can you form about the underlying rules or structure of the environment?
   Are there any actions that seem particularly promising or detrimental?
   Do certain sequences of actions lead to predictable outcomes?
- **Hypothesize**: Clearly state your current hypothesis about the most effective strategy for exploration or achieving a goal.
- **Plan**: Based on your reasoning and hypothesis, plan your next action or sequence of actions. Aim to test your hypothesis and gather more information.

{action_reward_description}

Respond with this format, please be specific about the object:

* Action: pick up <colored> <textured> <object>
* Stop: <YES> or <NO>
*
* Which combination of factors influence reward? <COLOR, SHAPE> or <COLOR, TEXTURE> or <TEXTURE, SHAPE> or <UNSURE>
* WINNING COMBINATION: <State the specific combination of properties (e.g., color and shape, shape and texture, or color and texture.>

Explain your reasoning thoroughly. Don't just guess! Each turn is precious. |

Table 2: In-context prompt used for the text environments for the conjunction tasks.

| Task | Prompt |
|---|---|
| Self-Correction | Task: You are tasked with exploring an environment efficiently.
You are given a description of the environment, a specific exploration goal,
and a proposed next step for exploration, along with the reasoning behind it.

Your Job:

Evaluate the proposed solution: Carefully analyze the proposed next step and its reasoning.
Consider whether it aligns with the overall exploration goal and efficiently gathers information
about the environment.

Identify errors: Determine if there are any flaws in the proposed solution's logic, efficiency,
or effectiveness in achieving the exploration goal.

Correct and improve: If you find errors, provide a corrected next step and explain your reasoning.
Your solution should be more effective or efficient than the proposed solution.

Accept if valid: If you find no errors in the proposed solution,
simply output the proposed solution and state that it is a valid approach.

TASK: {task}

SOLUTION: {solution} |

Table 3: Self-correction in-context prompts used for the text environment in the exploration phase for the Gemini agent.

| Task | Prompt |
|---|---|
| Guided reasoning | You are an AI agent designed for thoughtful exploration.

Your mission is to navigate and learn within a given environment by performing actions and observing the outcomes.

Operate as a scientist, carefully considering your actions and their consequences.

Exploration Cycle:

- **Action**: Choose an action to perform within the environment.

- **Observe**: Observe the result of your action. This includes any changes to the environment and any rewards or penalties received.

- **Record**: Maintain a detailed log of your actions, observations, and received rewards.

- **Review**: Periodically, pause to explicitly review your action history and the corresponding outcomes. Analyze this data to identify patterns, trends, and potential cause-and-effect relationships.

- **Reason**: Based on your review, reason about the environment.

If no reward has been received: Systematically explore new combinations of color, shape, and texture.
For example, if 'red', 'ball', and 'wood' have not been tried, pick a 'red wooden ball'.
If 'blue', 'cube' and 'steel' haven't been tried, pick a 'blue steel cube'.

If one or two objects with a reward have been found: Isolate the feature combinations causing the reward.
If a 'blue plastic cube' was rewarding, try a 'blue wooden cube' and a 'red plastic cube'
to see if the reward is related to the combination of color and texture, or shape and texture.
Continue this process until you have at least two objects with a reward of 1.
If more than two objects with a reward have been found: Explore randomly.

- **Hypothesize**: Clearly state your current hypothesis about the most effective strategy for exploration or achieving a goal.

- **Plan**: Based on your reasoning and hypothesis, plan your next action or sequence of actions.
Aim to test your hypothesis and gather more information. |

Table 4: Guided reasoning prompt used in Section 4.3 for the text environment in the exploration phase for the Gemini agent.

| Task | Prompt |
|---|---|
| 3D Environment Iterative Exploration: vision | You are an expert video game player who is annotating videos of gameplay.

In this game, the player controls a robot in a factory room, which contains objects of various shapes and colors, such as red planks, blue cubes, green cylinders, orange disks, yellow pyramids, etc.
The player can pick up and move objects using a blue laser beam.
The player is trying to place the correct type of object on the conveyor belt.
If the object is correct, the object disappears in the machine and the light on the machine turns green.
If the object is incorrect, the light on the machine turns red and the object is pushed off.

The possible colors are red, green, blue, yellow, purple, and orange.
The possible shapes are cylinder, cube, plank/board, pyramid, and disk.

Your goal is to accurately and comprehensively list every object that the player places on the input conveyor belt, along with the timestamp of when the object was placed and whether the object is correct or incorrect.

Your response should be in the following format:
0 [timestamp 0] <1st object placed on conveyor> : <correct / incorrect>
1 [timestamp 1] <2nd object placed on conveyor> : <correct / incorrect>
2 [timestamp 2] <3rd object placed on conveyor> : <correct / incorrect>
3 [timestamp 3] <4th object placed on conveyor> : <correct / incorrect>
... |
| 3D Environment Iterative Exploration: reasoning | Now we want to explain how this game works.
The goal of the game is to place all objects with the right property, such as a particular color or shape, on the conveyor belt.
Let's try to find the next action to take to figure out what factor (color or shape) determines the correctness of the object.

If there is no history of objects yet, tell the player to pick up a random object you can see in the room from the video.
If you have no video input yet, tell the player to explore the room.
Otherwise, follow the instructions below.

  Important: You have VERY FEW turns left. Choose your next action carefully to maximize information.

  Think step-by-step:

  1. What pattern do you see in the correct objects so far?
  2.  **Consider which colors and shapes have NEVER been correct. This eliminates BOTH the color AND shape from being correct.**
  3. What color or shape seems MOST promising to test next?
  4. Why will this choice give you the most useful information, even if it isn't a correct object?

  Explain your reasoning thoroughly. Don't just guess! Each turn is precious.

  After doing your reasoing, respond at the end with this format, please be specific about the object:

  * CORRECT PROPERTY: <COLOR> or <SHAPE> or <UNSURE>
  * NEXT COMMAND: place the <colored> <object> on the conveyor belt. |

Table 5: In-context prompts used for the 3D Construction Lab environment in the exploration phase for the Gemini agent.

| Task | Prompt |
|------|--------|
| 3D Environment Trajectory Review: vision | You are an expert video game player who is annotating videos of gameplay.

In this game, the player controls a robot in a factory room, which contains objects of
various shapes and colors, such as red planks, blue cubes, green cylinders,
orange disks, yellow pyramids, etc.
The player can pick up and move objects using a blue laser beam. The player
is trying to place the correct type of object on the conveyor belt. If the object
is correct, the object goes through and the light on the machine turns green.
If the object is incorrect, the light on the machine turns red and the object
is pushed off.

The possible colors are red, green, blue, yellow, purple, and orange.
The possible shapes are cylinder, cube, plank/board, pyramid, and disk.

Your goal is to accurately and comprehensively list every object that the
player places on the input conveyor belt, along with the timestamp of when the object was placed
and whether the object is correct or incorrect.

Your response should be in the following format:
0 [timestamp 0] <1st object placed on conveyor> : <correct / incorrect>
1 [timestamp 1] <2nd object placed on conveyor> : <correct / incorrect>
2 [timestamp 2] <3rd object placed on conveyor> : <correct / incorrect>
3 [timestamp 3] <4th object placed on conveyor> : <correct / incorrect>
... |
| 3D Environment Trajectory Review: reasoning | Now we want to explain how this game works.
The goal of the game is to place all objects with the right property, such as a particular color or shape,
on the conveyor belt.

Based on the observations above of which objects were placed on the conveyor belt
and which ones were correct or incorrect,  explain your reasoning and state what the right object
property is.
The right property is either a specific shape or a specific color.

Your response should be in the following format:
REASONING: <Explain your reasoning for how you deduced the right object property.>
TARGET PROPERTY: <State what the specific correct shape OR specific correct color is.> |
| 3D Environment Trajectory Review: generalization | Based on what you determined the correct object property to be, state whether
    each of the following objects would be correct if placed on the conveyor belt: |

Table 6: In-context prompts used for the 3D Construction Lab environment in the review phase for all agent conditions.