# OpenReview forum: "Can foundation models actively gather information in interactive environments to test hypotheses?"
_ICLR.cc/2025/Conference — ICLR 2025 Conference Withdrawn Submission_

### Official Review · Reviewer_58aK · 2024-10-30

**Soundness:** 4
**Presentation:** 3
**Contribution:** 2
**Rating:** 3
**Confidence:** 5

**Summary:**

The paper presents a study focused on evaluating the information-gathering abilities of foundation models within interactive environments. The authors introduce a novel framework designed to assess how these models can strategically gather and reason about information to solve problems. The framework is implemented in two distinct settings: a text-based environment and an embodied 3D simulation. The study specifically examines the performance of the Gemini 1.5 model in zero-shot settings, without task-specific training. The paper's key findings include a evaluation of the model's performance in the proposed benchmarks, a trade-off when multiple features must be identified concurrently, a comparison between different environments, and a detailed discussio of the analysis of the experiment results. Overall, the paper contributes a new framework for evaluating directed exploration capabilities, offers empirical analysis through extensive experiments.

**Strengths:**

Generally, the paper makes a strong case for the importance of information-gathering capabilities in foundation models and contributes valuable knowledge that can inform the development and application of AI systems. There are some strengths of the paper:
- The selected topic the researchers focuses on seems interesting. This framework allows for the evaluation of models' ability to strategically gather and reason about information in a systematic way.
- The implementation of the framework in both text-based and embodied 3D environments offers a broad perspective on the models' abilities, from controlled text interactions to more complex, real-world-like simulations.
- The discussion on the implications of the findings for future research and the development of autonomous intelligent agents provides a roadmap for further exploration and application of foundation models.

**Weaknesses:**

- The study primarily focuses on the Gemini 1.5 model, which may not fully represent the capabilities and behaviors of other foundation models. As a benchmark, evaluating a wider range of models could provide a more comprehensive understanding. This constraint limits the applicability and generalization of the study's findings to other models.
- From my point of view, the assessment of pure LLMs' strategic information-gathering abilities appears less meaningful (e.g., compared with RL agents) due to the agents' inability to engage directly in dialogue with the environment. This limitation hinders the ability to mimic human-like information-seeking behaviors through questioning and conversation. I think a more compelling focus could be evaluating agents powered by LLMs. While for agents, there are some pipelines designed for ensuring the information gathering in different environments.
- The experimental setups appear somewhat overly simplistic, lacking the complexity needed to truly challenge the models' capabilities. The distinction between text-based and embodied environments appears unnecessary. Is the primary difference the visual input versus text input? Also, I think the research would benefit from incorporating a greater variety of examples (which somewhat related to real-world applications) or expanding its applications to demonstrate a broader utility and impact.
- The paper may fall short in demonstrating substantial contributions, appearing to merely transpose traditional cognitive experiments to test large language models (LLMs) without significant innovation.

**Questions:**

See Weaknesses part.

---

### Official Review · Reviewer_1wtj · 2024-11-03

**Soundness:** 2
**Presentation:** 2
**Contribution:** 2
**Rating:** 3
**Confidence:** 5

**Summary:**

This paper explores the capability of foundation models to actively gather information for hypothesis testing in interactive environments. The authors propose a framework to evaluate these abilities in both a text-based environment and a embodied simulation (video input). Key findings include that while foundation models like Gemini 1.5 show near-optimal information-gathering efficiency in simple tasks, their performance decreases with increased task complexity, especially when conjunctions of features determine rewards. The study highlights challenges such as policy translation and in-context memory use, noting that visual inaccuracies in embodied environments further impact outcomes. The work concludes by identifying areas for improvement in visual and reasoning capabilities to enhance real-world application robustness.

**Strengths:**

- The paper is well-motivated, with a clear goal of studying active information gathering in large language models.

- The task is simple enough for addressing the scientific questions authors trying to ask, with a minimum amount of confounding factors.

- The paper is well-written and easy to read.

**Weaknesses:**

- Evaluations: the evaluations are not sufficient enough in many ways. First, while the paper is studying a human-like learning problem, but there is no human baseline presented. For example, would humans reach a near-optimal policy? Or they are more like the Gemini tested? Second, the title says foundations model"s". However, only the Gemini 1.5 model was tested. How do other models (Claude and GPTs) perform?

- Relations to prior works: the setting authors introduced is not new. I think several previous works have proposed similar environments: from simpler ones (simple object attributes corresponding to rewards [A, B, C] to complex causal rules [D]). None of these works are properly discussed in this paper.

- I'm suspecting on what broader implications the experiment results and discussions this paper can provide. So the answer the scientific question stated in the paper title is yes I guess?

[A]. Fränken, J. P., Theodoropoulos, N. C., & Bramley, N. R. (2022). Algorithms of adaptation in inductive inference. Cognitive Psychology, 137, 101506.

[B]. Xu, M., Jiang, G., Liang, W., Zhang, C., & Zhu, Y. (2024). Interactive visual reasoning under uncertainty. Advances in Neural Information Processing Systems, 36.

[C]. Kosoy, E., Chan, D. M., Liu, A., Collins, J., Kaufmann, B., Huang, S. H., ... & Gopnik, A. (2022). Towards understanding how machines can learn causal overhypotheses. arXiv preprint arXiv:2206.08353.

[D]. Wang, J. X., King, M., Porcel, N. P. M., Kurth-Nelson, Z., Zhu, T., Deck, C., ... & Botvinick, M. Alchemy: A benchmark and analysis toolkit for meta-reinforcement learning agents. In Thirty-fifth Conference on Neural Information Processing Systems Datasets and Benchmarks Track (Round 2).

**Questions:**

What's the role of the 3D version here? And meanwhile, I am curious about how curiosity-driven exploration agents perform here.

---

### Official Review · Reviewer_oj3H · 2024-11-04

**Soundness:** 3
**Presentation:** 4
**Contribution:** 2
**Rating:** 3
**Confidence:** 5

**Summary:**

This paper propose a parametric class of environments for testing LLM abilities to formulate hypotheses and interactively gather information. The environments vary in complexity, and come in a text-based and an embodied 3D implementation.
The evaluation is focused on Gemini in several model variants and prompting strategies, which is shown to outperform random baselines under various conditions.

**Strengths:**

This is a very well written paper with great figures (not a common occurrence in ICLR papers). Presentation is prefect. I enjoyed reading this paper.

Simple and well designed experimental task, which will be easy for a broad audience to understand.

**Weaknesses:**

Evaluation is focused on Gemini.

I feel like the paper is very well presented, but in terms of research questions illuminated by this paper it is going after a low-hanging fruit. It is super easy to implement a hypothesis testing task in a text prompt, and to compare to random/optimal information seeking baselines.

The main contribution of this paper seems to come from staging the contribution in a visual 3D task, however this staging does not tell us a lot about whether and how LLM explore.

**Questions:**

n/a

---

### Official Review · Reviewer_pRiX · 2024-11-04

**Soundness:** 3
**Presentation:** 2
**Contribution:** 2
**Rating:** 5
**Confidence:** 3

**Summary:**

The paper aims to assess the ability of foundation models to gather more information to finish a task/achieve a goal in interactive environments, with simple and complex reward functions.

In order to do so, the authors build a text-based and an embodied 3D environment where the goal is to find properties/concepts of objects (shape, color, etc.) which lead to maximum reward, without any prior information. The 3D environment only has visual cues, and thus provides a harder problem to solve. In 3D environment, Gemini only provides textual instructions to a human, who then performs it in the environment.

The agents are evaluated based on their efficiency (number of steps) required to gather the information needed to solve the problem, and their accuracy (how often the correct property is identified given fixed budget of steps).

The authors evaluate variants of Gemini model (Flash, 1.5 Pro) against random baselines (with and without replacement), and optimal baselines (a rule-based agent that optimizes for information gain). The task complexity varies in number of features in combination (single feature vs conjuction) that are to be identified as rewarding.

The learning/inference is designed as a two-stage process - vision and reasoning - where Gemini first lists down all the previous placements, timestamps, and rewards, and then tries to pick next action accordingly.

They show that Gemini models perform better than random baselines, and that total number of steps needed to find optimal properties increases with complexity.

They also experiment with self-correction, long-context windows and guided reasoning show that self-correction is more effective in complex conjunction tasks. The long-context window leads to improvement in conjunction tasks too. They also find that removing cases with visual errors leads to significant improvement in the 3D environment setting.

**Strengths:**

- The authors attempt to explore a novel aspect of foundation models - exploration in interactive environments, which is a unique study in its own.
- The baselines (upper/lower bounds) are logically sounds. I particularly like the idea of an optimal baseline to know where the most efficient agent would lie in the spectrum.
- The authors perform logical extensions (self-correction, long-context) and ablations (e.g. with correct visual outputs in the 3D environments) which help understand how well the agents perform with these added features.
- The writing is more or less clear, and the figures are helpful in understanding the paper better.

**Weaknesses:**

- The evaluation is limited - for some reason the authors only consider the Gemini-based agent, and not any other LLM models. Curious to hear an explanation on why this was done. What about the results on other LLM families?
- The results in Figure 3 are not unexpected. I would expect a larger LLM to be more efficient on account of more training/generalization ability, and LLMs to be worse than optimal baseline and better than random agents. I think there should be some more exploration towards finding new/unexpected insights or maybe a deeper analysis of the results.
- The environments are simple and the approach is not easily extensible to more real-world scenarios. I think disentanglement of the exploration component is important, however there should also be some thoughts on how to disentangle this component in more realistic tasks like navigation/more complex search-based problems, etc. Even the 3D environment has limited shapes/colors and limited number of correct objects.
- The downstream use-cases/impact of this work is not discussed, and is not immediately clear from the reading.
- Guided reasoning is not described in the main paper, but discussed. It also seems to achieve the best results in Fig 4 b.
- For the 3D environment, a human performs the instructions specified by Gemini. Can there ever be cases where Gemini instructions are vague? If not, there needs to be some discussion (if only a few lines) on this.
	- What kind of expertise does it need from the humans?
- "A likely reason for this is that the iterative nature ... occasional erros" -  What happens when the two-stage process is removed and a single step process is used i.e. direct reasoning from the previous data?  This seems like an important ablation.
- Minor:
	- Line 374: Guided reasoning is mentioned here, but not mentioned anywhere before.
	- Line 487: "aren't" -> are not.

**Questions:**

- Why are the baselines evaluated on 1k episodes, but Gemini on 200 episodes?
- Long-context does not seem to be helping in the single-feature case, what are the authors thoughts on this?
- Why do the authors think that Gemini 1.5 Flash performs better than Gemini 1.5 Pro on single-feature tasks?

---

### Note · Authors · 2024-11-27

I have read and agree with the venue's withdrawal policy on behalf of myself and my co-authors.